# DecoyDB: A Dataset for Graph Contrastive Learning in Protein-Ligand Binding Affinity Prediction

**Yupu Zhang**[1] **Zelin Xu**[1] **Tingsong Xiao**[1] **Gustavo Seabra**[2]
**Yanjun Li**[1,2] **Chenglong Li**[2] **Zhe Jiang**[1*]

[1]Department of CISE, University of Florida, Gainesville, FL, USA
[2]Department of Medicinal Chemistry, University of Florida, Gainesville, FL, USA
{y.zhang1, zelin.xu, xiaotingsong, yanjun.li, zhe.jiang}@ufl.edu
{seabra, lic}@cop.ufl.edu

## Abstract

Predicting the binding affinity of protein-ligand complexes plays a vital role in drug discovery. Unfortunately, progress has been hindered by the lack of large-scale and high-quality binding affinity labels. The widely used PDBbind dataset has fewer than 20K labeled complexes. Self-supervised learning, especially graph contrastive learning (GCL), provides a unique opportunity to break the barrier by pretraining graph neural network models based on vast unlabeled complexes and fine-tuning the models on much fewer labeled complexes. However, the problem faces unique challenges, including a lack of a comprehensive unlabeled dataset with well-defined positive/negative complex pairs and the need to design GCL algorithms that incorporate the unique characteristics of such data. To fill the gap, we propose DecoyDB[2], a large-scale, structure-aware dataset specifically designed for self-supervised GCL on protein–ligand complexes. DecoyDB consists of high-resolution ground truth complexes ($\leq$ 2.5Å) and diverse decoy structures with computationally generated binding poses that range from realistic to suboptimal. Each decoy is annotated with a Root Mean Square Deviation (RMSD) from the native pose. We further design a customized GCL framework to pretrain graph neural networks based on DecoyDB and fine-tune the models with labels from PDBbind. Extensive experiments confirm that models pretrained with DecoyDB achieve superior accuracy, sample efficiency, and generalizability.

## 1 Introduction

Drug discovery is a lengthy and costly endeavor that involves identifying highly potent molecules capable of interacting with specific molecular targets, such as proteins, to treat diseases. Predicting protein–ligand binding affinity is a fundamental task in the early stages of drug discovery [Kitchen *et al.*, 2004], with two major applications. First, it enables virtual screening and *de novo* design—key computer-aided drug discovery techniques—to efficiently narrow down promising hit compounds from a vast chemical space targeting a specific protein [Li *et al.*, 2022b]. Second, it supports the subsequent optimization of these identified molecules to further improve their binding affinities and other pharmacological properties [Li *et al.*, 2020].

Classical methods typically employ theory-inspired, fixed functional forms that utilize predefined features extracted from the protein–ligand complex to estimate affinity. Although certain computational techniques such as molecular mechanics/Poisson–Boltzmann or generalized Born surface

---

*Corresponding author. Email: zhe.jiang@ufl.edu
[2]Code and data are available at: huggingface.co/datasets/jiangteam/DecoyDB.

area (MM/PBSA, MM/GBSA) [Genheden and Ryde, 2015], thermodynamic integration (TI) [Kirkwood, 1935], free energy perturbation (FEP) [Zwanzig, 1954], can offer decent precision, they are computationally intensive and impractical for use in virtual screening.

In recent years, deep learning-based affinity prediction has seen significant improvement [Rezaei *et al.*, 2022; Zhang *et al.*, 2024], with models utilizing 3D convolutional neural networks (CNNs) [Stepniewska-Dziubinska *et al.*, 2018b; Li *et al.*, 2019] and graph neural networks (GNNs) [Lim *et al.*, 2019; Yang *et al.*, 2023] to learn structural representation from ligands and proteins. Despite these advances, the accuracy of current scoring functions remains unsatisfactory, and further improvements are hindered by the limited availability of ground truth protein-ligand binding affinity labels. For instance, the widely used PDBbind dataset [Wang *et al.*, 2005] contains fewer than 20,000 labeled complexes. Such labels are typically obtained through laborious and time-consuming experiments and thus are unlikely to grow dramatically. In contrast, there is an abundance of unlabeled protein-ligand complexes in structural databases, which could potentially enhance the learning of spatial structural interaction features for binding affinity prediction.

Self-supervised learning (e.g., masked autoencoders, contrastive learning) has shown great promise for pretraining deep neural networks on vast amounts of unlabeled data, followed by fine-tuning with limited labeled samples for diverse downstream tasks such as natural language processing [Radford *et al.*, 2018] and computer vision [He *et al.*, 2022]. Among these approaches, contrastive learning has emerged as a particularly effective framework for learning representations from unlabeled graph data [He *et al.*, 2020; Chen *et al.*, 2020], including protein–ligand complexes [Luo *et al.*, 2024]. Motivated by this success, we adopt contrastive learning as a pretraining strategy to leverage large-scale unlabeled protein–ligand complexes, enabling subsequent fine-tuning on smaller labeled datasets for accurate binding affinity prediction.

However, several unique challenges arise in this context. First, there is a lack of a comprehensive unlabeled dataset containing well-defined positive and negative complex pairs, which are essential for contrastive learning. Second, preserving the original 3D structural integrity of protein–ligand complexes is critical for maintaining biochemical validity. Conventional graph contrastive learning techniques—such as generating augmented pairs through node or edge perturbations [Chen *et al.*, 2020; Wang and Qi, 2022]—may produce unrealistic conformations that violate physical and chemical constraints [Qin *et al.*, 2024]. Third, the downstream task of binding affinity prediction requires the model to capture interactions corresponding to low-energy (stable) conformations, a property that standard contrastive learning objectives typically overlook [Liu *et al.*, 2023].

To fill this gap, we propose *DecoyDB*, a comprehensive dataset of high-resolution, ground-truth 3D protein–ligand complexes filtered from the PDB, augmented with diverse decoy complexes generated computationally to produce binding poses ranging from realistic (positive pairs) to suboptimal (negative pairs). *DecoyDB* contains 61,104 ground-truth 3D complexes and 5,353,307 decoys, and each decoy is annotated with its Root Mean Square Deviation (**RMSD**) from the native pose, i.e., the spatial distances between the atoms of the decoy's ligand and the corresponding atoms in the original ligand. We further design a customized graph contrastive learning algorithm that incorporates (1) a two-category contrastive loss, where negative samples are drawn both from decoys of the same complex (with varying RMSD levels) and from different real complexes, and (2) a denoising score matching (DSM)-based regularization term in the loss. Extensive experiments demonstrate that our pretraining framework improves base models in prediction accuracy, sample efficiency, and generalizability. It is worth noting that although we focus on enhancing binding affinity prediction, *DecoyDB* also has potential for broader applications, such as molecular docking and virtual screening.

## 2 Related Work

**Deep Learning for Protein-Ligand Binding Affinity Prediction:** In recent years, deep learning has emerged as a powerful tool for predicting protein-ligand binding affinity. Earlier methods are mostly based on CNNs, including Pafnucy [Stepniewska-Dziubinska *et al.*, 2018b], OnionNet [Zheng *et al.*, 2019], and DeepAtom [Li *et al.*, 2019], which typically rasterize the binding pocket and ligand into a grid structure. However, these methods are inherently dependent on the grid resolution and do not consider the topological structure of a complex. In addition, some methods for drug target affinity prediction, such as DeepDTA [Lennox *et al.*, 2021] and GraphDTA [Nguyen *et al.*, 2020], miss the interaction structures. Recent works have shifted towards graph neural networks (GNNs) to

Table 1: Public datasets related to protein–ligand complexes for binding affinity prediction.

| Category | Dataset | # of complexes | 3D structure | Affinity | Measurement |
|---|---|---|---|---|---|
| Both affinity labels and 3D structure | PDBbind2013 | 10,370 | Exp. | Exp. | $IC_{50}, K_d, K_i$ |
| | PDBbind2016 | 13,189 | Exp. | Exp. | $IC_{50}, K_d, K_i$ |
| | PDBbind2020 | 19,443 | Exp. | Exp. | $IC_{50}, K_d, K_i$ |
| | MISATO | 19,443 | Exp.+QC+MD | Exp. | $IC_{50}, K_d, K_i$ |
| | LP-PDBbind | 18,795 | Exp. | Exp. | $IC_{50}, K_d, K_i$ |
| | BDB2020+ | 115 | Exp. | Exp. | $IC_{50}, K_d, K_i$ |
| | HiQBind | 32,275 | Exp. + Refined | Exp. | $IC_{50}, K_d, K_i$ |
| | Binding MOAD | 41,409 | Exp. | Exp. | $IC_{50}, K_d, K_i$ |
| | BioLip | 48,291 | Exp. | Exp. | $IC_{50}, K_d, K_i$ |
| | BindingNet | 69,816 | Exp.+Comp. | Exp. | $IC_{50}, K_d, K_i$ |
| Only affinity labels, no 3D structure | KIBA | 117,657 | - | Exp. | KIBA ($IC_{50}, K_d, K_i$) |
| | Davis | 30,056 | - | Exp. | $K_d$ |
| | BindingDB | 679,000 | - | Exp. | $IC_{50}, K_d, K_i$ |
| No affinity labels, only 3D structure | PDB (05/2025) | 178,900 | Exp. | - | - |
| | Redocked 2020 | 786,960 | Exp. + Decoys | - | - |
| | DecoyDB | 5,414,411 | Exp. + Decoys | - | - |

learn a flexible representation of 3D graph structures, such as PotentialNet [Feinberg *et al.*, 2018], IGN [Jiang *et al.*, 2021], EGNN [Satorras *et al.*, 2021], PSICHIC [Koh *et al.*, 2024], and GIGN [Yang *et al.*, 2023].

**Graph Contrastive Learning:** Generic graph contrastive learning (GCL) methods generate positive and negative sample pairs through random node dropping, perturbation, or subgraph sampling [You *et al.*, 2020; Xu *et al.*, 2021; Tong *et al.*, 2021], which often disrupt biochemical properties. Although recent advances have introduced GCL frameworks specifically for molecular data [Guan and Zhang, 2023; Fang *et al.*, 2023; Liu *et al.*, 2022; Sun *et al.*, 2021; Li *et al.*, 2022a], these models typically focus on a single molecular graph and fail to capture the interaction network between a protein and its ligand. Wu *et al.* [2022] proposed a self-supervised learning method but required additional molecular dynamics simulations. Ni *et al.* [2024] also evaluated their molecular foundation model on protein–ligand binding affinity prediction, but their base model was pretrained only on single molecules. Luo *et al.* [2024] introduced a two-step supervised learning framework for protein–ligand binding prediction, consisting of (1) a supervised learning step with an auxiliary contrastive loss based on decoys, and (2) a supervised parameter refinement step using true complex affinity labels only. However, this approach is not true pretraining, as the first step's loss includes pseudo-labels of binding affinity for decoys, and their decoy dataset was derived solely from PDBbind (relatively small in size). In addition, their contrastive loss does not account for varying degrees of negativeness among decoys with different deviations, and the anchors in their triplets are complex-free (lacking interaction structural information).

**Protein-ligand complex datasets:** Table 1 summarizes the existing public protein-ligand complex datasets. Most datasets are derived from the Protein Data Bank (PDB) [Berman *et al.*, 2000], a comprehensive repository of experimentally determined 3D biomolecular structures. PDB (as of 05/2025) provides around 178,900 protein-ligand complexes but does not provide binding affinity labels. To fill the gap, PDBbind provides fewer than 20K complexes with experimentally measured affinity and is the most widely used benchmark for binding affinity prediction. Several datasets aim to enrich or refine the PDBbind with additional structural or physical information. For example, MISATO [Siebenmorgen *et al.*, 2024] augments PDBbind entries with quantum-chemically (QC) optimized ligand geometries and 10-nanosecond molecular dynamics (MD) trajectories. LP-PDBbind [Li *et al.*, 2024a] provides a leakage-proof split of PDBbind based on structural information. In addition, the authors construct a high-quality test set, BDB2020+, by selecting new protein-ligand complexes published after PDBbind2020 for independent testing. There are several additional labeled datasets. Binding MOAD [Wagle *et al.*, 2023] provides 41,409 complexes, of which only 15,223 (37%) are annotated with affinity labels. BioLiP [Yang *et al.*, 2012] integrates Binding MOAD, PDBbind, and BindingDB, providing over 48,000 labeled complexes, but it does not apply resolution filtering and thus can contain low-quality 3D complex structures. HiQBind [Wang *et al.*, 2025] collects 32,275 protein-ligand complexes with refined structures. Despite their high-quality affinity labels, these datasets remain relatively small in scale, and there exist substantial overlaps between them. For example, BioLiP shares 26,009 complexes with Binding MOAD, and most complex structures in HiQBind are from Binding MOAD and BioLiP. To support larger-scale labels, BindingNet [Li *et al.*, 2024b] expands PDBbind to 69,816 complexes by superimposing ligands in the original complexes.

However, since the 3D complex structures are computationally generated rather than experimentally resolved, discrepancies may exist between the modeled conformations and their corresponding binding affinities. Separately, several other datasets such as KIBA [Tang *et al.*, 2014], Davis [Davis *et al.*, 2011], and BindingDB [Liu *et al.*, 2007] provide a larger number of binding affinity labels, but they do not provide corresponding 3D conformal structures. In summary, most existing datasets are designed exclusively for supervised learning and therefore cannot leverage the vast number of unlabeled 3D complexes. One existing dataset, Redocked2020 [Francoeur *et al.*, 2020], augments 3D complexes in PDBbind to 786,960 binding poses by adding decoys (via redocking ligands into protein structures) for self-supervised pretraining based on graph contrastive learning [Luo *et al.*, 2024]. However, since the 3D complexes are entirely derived from PDBbind, there exists leakage between pretraining and fine-tuning. In contrast, our DecoyDB is filtered from PDB with many complexes outside the PDBbind dataset, providing an opportunity to validate the generalizability of pretrained models during fine-tuning.

## 3 Problem Definition

**Definition 1.** *A **protein–ligand complex** refers to the binding interaction between a protein and a ligand. It can be represented as $s_k = (G_k, \mathbf{x}_k, \mathbf{a}_k)$, where $G_k = (V_k, E_k)$ is a 3D graph whose nodes ($v \in V_k$) represent atoms of the protein or ligand, and whose edges ($e \in E_k$) represent chemical interactions either within the protein, within the ligand, or between them (interaction edges). $\mathbf{x}_k \in \mathbb{R}^{|V_k| \times l}$ denotes the matrix of spatial node features ($l = 3$ for 3D coordinates). $\mathbf{a}_k \in \mathbb{R}^{|V_k| \times m}$ denotes the matrix of non-spatial node features (e.g., atom type or other chemical descriptors), where $m$ is the number of features. $y_k \in \mathbb{R}$ is the binding affinity score associated with the complex, which is unavailable in an unlabeled dataset.*

The problem of protein-ligand binding affinity prediction can be formally defined as follows:

**Input:**
• A set of protein-ligand complexes $\mathcal{S} = \{s_k | 1 \le k \le K\}$.
• A small subset of ground truth binding affinity scores $\mathcal{Y} = \{y_k | 1 \le k \le K_l\}$, where $K_l << K$.
• A base GNN encoder $f_\theta$ and regression head $g_\phi$: $\hat{y}_k = g_\phi(f_\theta(s_k))$
**Output:**
• Parameters $\theta$ of pretrained GNN encoder $f_\theta$ based on $\mathcal{S}$.
• Parameters $\phi$ in regression head $g_\phi$ based on $\mathcal{Y}$.
**Objective:**
• Minimize the prediction errors of the fine-tuned model.
• Maximize the label efficiency in fine-tuning.

## 4 Our *DecoyDB* Dataset

### 4.1 The Construction of *DecoyDB*

In order to use unlabeled complexes in self-supervised learning (graph contrastive learning), we need to first establish positive and negative sample pairs. Common data augmentation methods in general graph contrastive learning (e.g., edge perturbation, node dropping) are inadequate since they destroy biochemical structures. Thus, we constructed a specialized dataset to augment the original protein-ligand complexes with decoys. A decoy refers to an artificial protein-ligand complex with a computationally generated suboptimal binding pose. We name our augmented dataset *DecoyDB*.

Figure 1 illustrates our data construction process. We started by retrieving all available structures containing ligand-protein complexes from the PDB, focusing on entries obtained by X-ray crystallography with a resolution of 2.5Å or less. We refined our *DecoyDB* dataset by excluding complexes with ligands that had molecular weights outside the $(50, 700)$ range, metal clusters, monoatomic ions, common crystallization molecules, and ligands containing elements other than C (carbon), N (nitrogen), O (oxygen), H (hydrogen), S (sulfur), P (phosphorus), or X (halogens). To avoid redundancy and irrelevant protein chains, we isolated the target ligand in each remaining PDB entry and retained only those protein chains with at least one atom within 10Å of the ligand and saved a PDB file containing only the protein and ligand. This threshold captures relevant interactions, while excluding distant, noninteracting parts of the protein. In cases where multiple ligands were present

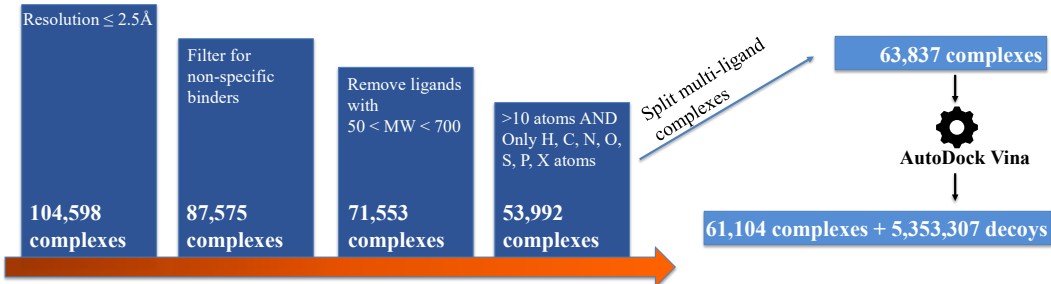

Figure 1: The data construction pipeline of DecoyDB.

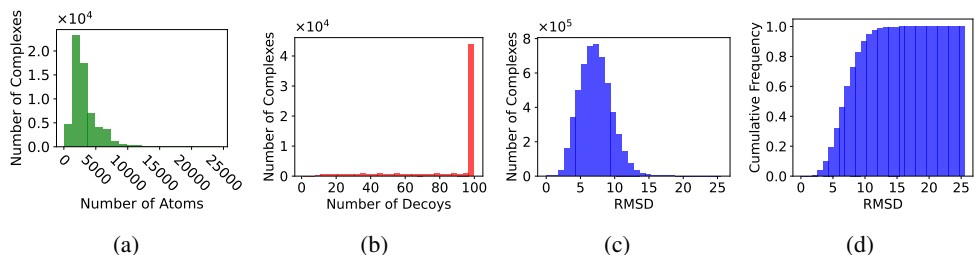

Figure 2: Statistical analysis of *DecoyDB*. (a) Distribution of the number of atoms in each protein-ligand complex. (b) Distribution of the number of decoys per complex. (c) Distribution of RMSD values for decoy complexes. (d) Cumulative distribution of RMSD values for decoy complexes.

within a complex, we generated a separate PDB file for each ligand-protein pair. We collected 63,837 3D complexes after the split.

To generate decoy complexes, we used AutoDock Vina 1.2 [Eberhardt *et al.*, 2021], one of the most widely used open-source programs for molecular docking. For each PDB file (a real protein-ligand complex), we defined a grid box around the ligand with a 5Å padding in each dimension to ensure sufficient space for ligand flexibility during redocking. We generated 100 different poses per ligand using an exhaustiveness parameter of 8, which balances computational efficiency with pose diversity. The final (RMSD) between the generated pose and the original crystallized ligand was calculated to quantify the deviation of the decoys from the native binding pose.

The resulting *DecoyDB* dataset contains 61,104 protein-ligand complexes and 5,353,307 decoys, with an average of 88 successfully generated decoys per complex. The data distribution is shown in Figure 2. The RMSD of the decoys ranges from 0.03Å to 25.56Å, with an average of 7.22Å. This wide range of RMSD values ensures a diverse set of decoys that represent both near-native (**positive samples**, RMSD $\leq$ 2Å) and far-from-native poses (**negative samples**, RMSD > 2Å), with the 2 Å cutoff following docking success criteria [Buttenschoen *et al.*, 2024; Cole *et al.*, 2005]. Such diversity is crucial for contrastive learning, as it allows the model to learn from a broad range of binding interactions and spatial conformations, improving its generalization in predicting protein-ligand binding affinity.

### 4.2 A Customized Graph Contrastive Learning Framework

We now introduce a customized graph contrastive learning framework that pretrains a neural network based on two-category negative sample pairs and denoising-based regularization.

**Two-category Graph Contrastive Loss with Continuous Negative Samples**

*DecoyDB* augments the original real complexes with positive and negative sample pairs. For each real complex (anchor), we define positive pairs by matching the complex with its decoys whose RMSD

is no greater than 2Å (positive samples), and define negative pairs in two categories: (1) matching the anchor complex with its decoys whose RMSD values is greater than 2Å), and (2) matching the anchor complex with a different complex. Based on the two-category negative pairs, we customize the graph contrastive loss.

Our proposed two-category InfoNCE contrastive loss is in Equation 1, where $\mathbf{z_k}$ is the embedding of a real complex (the *anchor*) from the encoder, and $\mathbf{z_i}$ is the embedding of one positive sample, $\mathbf{z_j}$ is the embedding of a negative sample, and $N$ is the total number of samples in the minibatch. Each minibatch has one positive pair and $N - 1$ negative pairs.

$$l_{k,i} = -\log \frac{\exp(sim(\mathbf{z_k}, \mathbf{z_i})/\tau)}{\sum_{j=1, j \neq i}^{N} \beta_{k,j} \exp\left(sim(\mathbf{z_k}, \mathbf{z_j})/\tau\right)} \tag{1}$$

One important factor in the above loss is $\beta_{k,j}$. It is defined in Equation 2, where $d_{\mathbf{z_k}, \mathbf{z_j}}$ is the RMSD between embeddings $\mathbf{z_k}$ and $\mathbf{z_j}$, $d_{max}$ is the max RMSD between the real complexes and their decoys (to normalize $d_{\mathbf{z_k}, \mathbf{z_j}}$), $sim$ is the cosine similarity function, and $\alpha$ is a hyper-parameter to balance the relative weight between two categories of negative samples. Specifically, if the negative sample $\mathbf{z_j}$ is from a decoy, we use a normalized continuous RMSD to reflect the extent of negativeness in the contrastive loss. For decoys that have more deviations from the anchor complex, our intuition is that their embeddings need to be pushed away further from the anchor. If the negative sample $\mathbf{z_j}$ is from a different complex (than the current anchor complex), then we use a binary weight $\beta = 1$.

$$\beta_{k,j} = \begin{cases} \alpha \frac{d_{\mathbf{z_k}, \mathbf{z_j}}}{d_{max}} & \text{if } \mathbf{z_j} \text{ is a negative decoy of } \mathbf{z_k}, \\ 1 & \text{otherwise.} \end{cases} \tag{2}$$

Suppose there are $K$ anchors in the complex dataset and each anchor has $m$ positive pairs. The loss function $L_1$ is a sum over all complexes and their positive pairs:

$$L_1 = \frac{1}{m \times K} \sum_{k=1}^{K} \sum_{i=1}^{m} l_{k,i} \tag{3}$$

**Denoising-based Regularization**

Although the above two-category graph contrastive loss helps supervise the relative closeness of embedding pairs, it does not reflect the fact that each real complex has a binding pose with minimum energy. Inspired by a recent work [Jin *et al.*, 2024], we add a denoising-based regularization method. Our intuition is that the real complex structure is the most stable structure and represents the local minimum potential energy in its pose.

Specifically, we add Gaussian noise to the 3D coordinates of atoms in the complex to get perturbed complex $s' = (G, \mathbf{x}', \mathbf{a})$, i.e., $\mathbf{x}' = \mathbf{x} + \epsilon$, where $\epsilon \sim p(\epsilon) = \mathcal{N}(0, \sigma^2 I)$, $\mathbf{x}$ are the coordinates of the atoms and $\mathbf{x}'$ are perturbed coordinates of atoms. Note that we add Gaussian noise to the ligand atoms only. We apply DSM only to original complexes (not decoys). For simplicity, we reuse the same symbol of $\mathbf{x}$ to represent the coordinates of a ligand. The model's output denotes the structural energy. Our intuition is that the gradient of the model's score with respect to the input vanishes in the absence of noise. Specifically, **denoising score matching (DSM)** loss based on [Zaidi *et al.*, 2023] is:

$$L_2 = \mathbb{E}_{q_\sigma(\mathbf{x}', \mathbf{x})} \left[ \frac{\partial \log f(\mathbf{s}')}{\partial \mathbf{x}'} - \frac{\mathbf{x} - \mathbf{x}'}{\sigma^2} \right] \tag{4}$$

where $q_\sigma(\mathbf{x}', \mathbf{x})$ is the joint distribution of perturbed and non-perturbed structures.

To optimize the performance on the prediction task, we integrate contrastive learning with noise-based regularization by combining two different constraints to formulate the overall objective function:

$$L = L_1 + \mu L_2 \tag{5}$$

Here, $\mu$ is a hyper-parameter adjusted to balance the contribution of contrastive learning with DSM-based regularization.

# 5 Experiments

**Dataset description:** For the pretraining phase, we used the proposed *DecoyDB* dataset and removed overlapping samples from fine-tuning binding affinity datasets. For other pretraining methods, Frad and ConBAP, we used the pretraining model provided by the authors. Specifically, Frad was pretrained on PCQM4Mv2 [Nakata and Shimazaki, 2017] and ConBAP was pretrained on Redocked2020 [Francoeur *et al.*, 2020]. For fine-tuning, we used the PDBbind2016 and PDBbind2013 datasets [Wang *et al.*, 2005], which contain 13,189 complexes and 10,370 complexes, respectively. Each sample is associated with a binding affinity. Note that we conduct pretraining only on GNN-based models. For other models, we train them directly on the PDBbind dataset without self-supervised pretraining. To ensure a fair comparison during fine-tuning, we followed the established setup by randomly selecting 11,904 training and 1,000 validation complexes for PDBbind2016, and 7,977 training and 1,000 validation complexes for PDBbind2013. For model testing, we used two independent benchmark test sets: the PDBbind2013 core set and PDBbind2016 core set, containing 107 and 285 complexes, respectively. These test sets are also removed from our pretraining dataset to conduct a rigorous evaluation of our pretraining framework.

**Evaluation metrics:** During pretraining, we used the early stopping strategy based on training and validation loss. During fine-tuning, we assessed the model's test performance using Root Mean Square Error (RMSE) and Pearson's correlation coefficient (R), following previous works [Stepniewska-Dziubinska *et al.*, 2018a; Zheng *et al.*, 2019].

**Baselines:** The baseline models include: Docking method (**AutoDock Vina** [Eberhardt *et al.*, 2021]), Drug-Target Affinity methods (**DeepDTA** [Lennox *et al.*, 2021], **GraphDTA** [Nguyen *et al.*, 2020]), CNN-based methods (**Pafnucy** [Stepniewska-Dziubinska *et al.*, 2018b], **OnionNet** [Zheng *et al.*, 2019]), GNN-based methods (**SchNet** [Schütt *et al.*, 2017], **EGNN** [Satorras *et al.*, 2021], **GIGN** [Yang *et al.*, 2023]), General GCL with edge perturbation on GIGN (**GIGN + GCL-EP**), GCL with node dropping on GIGN (**GIGN + GCL-ND**) and **pretraining for biochemistry graphs** (**Frad** [Ni *et al.*, 2024], **ConBAP** [Luo *et al.*, 2024]). We compare our customized GCL algorithm (named **OURS**) on DecoyDB against baselines. More details are provided in Appendix B.

**Implementation details:** For baseline models, we used the original source code provided by the authors. For the pretraining methods, we used the pretrained model parameters provided by the authors and fine-tuned them on our dataset, where ConBAP used EGNN as its base model, and Frad used TorchMD-Net [Thölke and De Fabritiis, 2022] as its base model. Our pretraining framework used these GNNs as the base encoders for complexes, with two separate dense layers during the pretraining and fine-tuning phases to make predictions. We used GIGN as the base model for the ablation study, sensitivity analysis, the impact of dataset size and the model generalization.

We used the same learning rate (5e-4) and weight decay (1e-6) for each model in two phases. During the pretraining phase, we trained each model for 20 epochs (Figure 4 (a)). In the fine-tuning phase, each model was trained for a maximum of 300 epochs, with early stopping applied if there was no improvement on the validation set within 40 epochs. The detailed setup can be found in our supplementary materials. All experiments were executed on an NVIDIA DGX-2 node with AMD EPYC 7742 64-core CPU and eight A100 GPUs. We ran each model ten times to obtain the average performance and standard deviations. Additional experimental details are provided in Appendix A.

## 5.1 Overall Performance

Table 2 summarizes the detailed comparison of our framework against baseline models on two datasets. Among the baselines, DTA-based methods show relatively high RMSE values, likely because they do not take into account the detailed spatial structures related to binding interactions. CNN-based models show moderate improvements, with reduced RMSE and increased R values, but their reliance on volume-based representations cannot capture the graph structural information. In contrast, GNN-based models, such as EGNN and GIGN, outperform other baselines, even without pretraining. Particularly, GIGN achieves the best accuracy among all base models. We also compared our pretraining framework with alternative contrastive learning methods (when their source codes were available) on several GNN base models. Results show that adding our pretraining framework significantly improves the base model performance. For instance, on the EGNN base model, our pretraining framework (OURS) reduces the RMSE from 1.304 to 1.250 on the PDBbind 2016, while the existing ConBAP method (which also uses decoys to augment negative samples) only reduces

Table 2: Performance comparison on PDBbind core set 2013 and PDBbind core set 2016.

| | Method | Pretraining dataset | PDBbind core set 2013 | | PDBbind core set 2016 | |
|---|---|---|---|---|---|---|
| | | | RMSE ↓ | R ↑ | RMSE ↓ | R ↑ |
| Docking | AutoDock Vina | - | 2.400 | 0.570 | 2.350 | 0.600 |
| DTA | DeepDTA | - | 1.603 (0.014) | 0.717 (0.016) | 1.366 (0.011) | 0.777 (0.013) |
| | GraphDTA | - | 1.742 (0.039) | 0.673 (0.032) | 1.543 (0.033) | 0.707 (0.021) |
| CNN | Pafnucy | - | 1.544 (0.024) | 0.778 (0.013) | 1.423 (0.040) | 0.793 (0.023) |
| | OnionNet | - | 1.562 (0.071) | 0.747 (0.031) | 1.421 (0.069) | 0.772 (0.024) |
| GNN +pretrain | TorchMD-Net | - | 1.466 (0.034) | 0.763 (0.016) | 1.294 (0.026) | 0.808 (0.011) |
| | +Frad | PCQM4Mv2 | 1.447 (0.016) | 0.780 (0.008) | 1.264 (0.043) | 0.811 (0.006) |
| | SchNet | - | 1.642 (0.030) | 0.739 (0.016) | 1.526 (0.037) | 0.744 (0.014) |
| | **+OURS** | **DecoyDB** | **1.577 (0.034)** | **0.763 (0.015)** | **1.481 (0.031)** | **0.755 (0.012)** |
| | EGNN | - | 1.496 (0.047) | 0.761 (0.012) | 1.334 (0.024) | 0.801 (0.010) |
| | +ConBAP | Redocked 2020 | 1.479 (0.038) | 0.766 (0.010) | 1.300 (0.019) | 0.802 (0.014) |
| | **+OURS** | **DecoyDB** | **1.437 (0.044)** | **0.781 (0.013)** | **1.267 (0.021)** | **0.813 (0.011)** |
| | GIGN | - | 1.421 (0.038) | 0.786 (0.016) | 1.262 (0.032) | 0.811 (0.010) |
| | + GCL-EP | **DecoyDB** | 1.417 (0.034) | 0.789 (0.014) | 1.251 (0.029) | 0.814 (0.008) |
| | + GCL-ND | **DecoyDB** | 1.420 (0.026) | 0.787 (0.011) | 1.254 (0.026) | 0.813 (0.013) |
| | **+OURS** | **DecoyDB** | **1.377 (0.039)** | **0.813 (0.013)** | **1.189 (0.031)** | **0.838 (0.011)** |

Table 3: P-values for pairwise comparisons between OURS and baselines on PDBbind 2013/2016. **Bold** denotes $p < 0.05$ (statistically significant).

| Comparison | RMSE (2013) | R (2013) | RMSE (2016) | R (2016) |
|---|---|---|---|---|
| SchNet + OURS vs SchNet | **0.00056** | **0.0038** | **0.00015** | 0.069 |
| EGNN + OURS vs EGNN | **0.040** | **0.00069** | **0.00086** | **0.034** |
| EGNN + OURS vs EGNN + ConBAP | 0.071 | **0.00043** | **0.045** | **0.022** |
| GIGN + OURS vs GIGN | **0.015** | **0.0014** | **0.00018** | **0.000046** |
| GIGN + OURS vs GIGN + GCL-EP | **0.019** | **0.0046** | **0.00012** | **0.00065** |
| GIGN + OURS vs GIGN + GCL-ND | **0.011** | **0.0032** | **0.00014** | **0.00036** |

the RMSE to 1.285. As another example, our pretraining method reduces the RMSE of GIGN (the best base model) from 1.460 to 1.386 on the PDBbind core set 2013 and from 1.263 to 1.188 on the PDBbind core set 2016. In contrast, general GCL does not show significant improvement on GIGN. For key comparisons, we also conducted paired t-tests on both RMSE and $R$. As summarized in Table 3, most improvements are statistically significant (p < 0.05). For the two borderline cases, we provide per-run results over ten runs in Appendix C.

## 5.2 Sensitivity Analysis

In this section, we evaluate the impact of two key hyperparameters $\alpha$ and $\mu$. $\alpha$ (Equation 2) influences the relative weight of the factor $\beta$ (i.e., relative weight of decoy negative samples versus negative samples from different complexes), while $\mu$ balances the relative weight of contrastive learning and DSM. We varied both parameters ranging from 0.4 to 2.0 and observed variations in RMSE across the test datasets. As shown in Figure 3, RMSE initially decreases and then increases with rising $\alpha$, but it is persistently lower than the RMSE of the baseline GIGN model (shown as the red dashed line). The results of $\mu$ follow a similar trend.

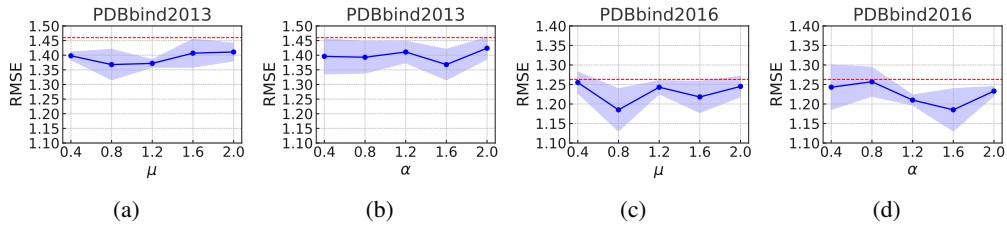

| (a) | (b) | (c) | (d) |

Figure 3: Sensitivity analysis of two key hyper-parameters, $\alpha$ and $\mu$, in our loss function, as shown in Equation 2 and Equation 5. (a) and (c) show the RMSE performance across different values of $\mu$ on the two datasets, PDBbind2013 and PDBbind2016. (b) and (d) show the RMSE variation with different values of $\alpha$ on the same two datasets. The red dashed lines show the baseline GIGN model.

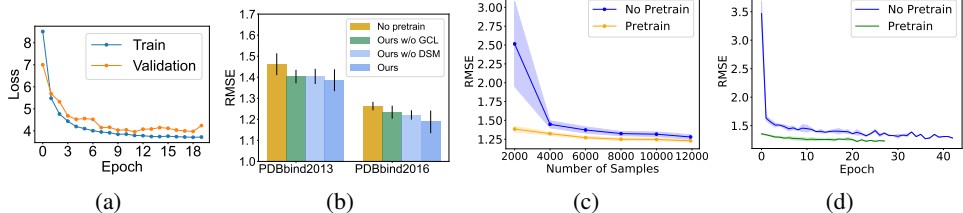

(a)       (b)       (c)       (d)

Figure 4: (a) is the training and validation loss curve during pretraining. (b) is the ablation study. (c) is the impact of fine-tuning dataset size on the binding affinity prediction. (d) is the validation curve in fine-tuning.

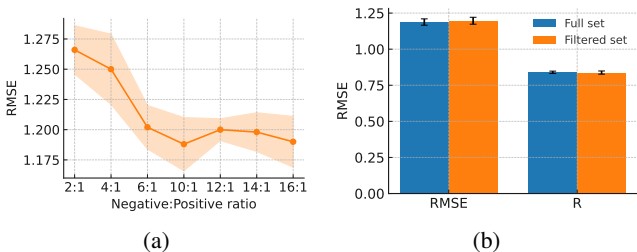

(a)                 (b)

Figure 5: (a) Negative to Positive ratio vs. RMSE; (b) Filtering out top 1% highest RMSD decoys.

## 5.3 Ablation Study

Table 4: Ablation study of our two-category graph contrastive loss.

| Method | PDBbind core set 2013 | PDBbind core set 2016 |
|---|---|---|
| with one-category graph contrastive loss | 1.406 (0.047) | 1.246 (0.014) |
| with two-category graph contrastive loss | **1.394** (0.036) | **1.221** (0.022) |

Table 5: Paired t-tests for ablations.

| Ablation comparison | p-value |
|---|---|
| No pretraining vs. OURS | 0.0280 |
| OURS w/o DSM vs. OURS | 0.0013 |
| OURS w/o GCL vs. OURS | 0.0325 |

Our ablation study evaluates the impact of two key modules, i.e., the graph contrastive learning (GCL) and denoising score matching (DSM)-based regularization in the pretraining phase. We used GIGN as the base model, starting with a full framework that included both components and then removing each component to observe the impact on model performance. As shown in Figure 4(d), removing either GCL or DSM led to a modest increase in RMSE (although their RMSE levels are still lower than the RMSE of no pretraining), indicating that both of them play a crucial role in capturing meaningful representations and improving performance. We also added an ablation study of our two-category graph contrastive loss. We pretrained two models, one model with two-category contrastive loss (both decoy negative samples and different complexes as negative samples, denoted as "with two-category graph contrastive loss") and the other with one-category GCL loss (only different complexes as negative samples, denoted as "without decoy samples"). Results in Table 4 show that the proposed two-category graph contrastive loss slightly outperforms a one-category contrastive loss on both datasets. To analyze each component's contribution, we run 10 times per variant and conduct paired two-sided t-tests on RMSE. As summarized in Table 5, all comparisons are statistically significant ($p < 0.05$), indicating that pretraining, DSM, and GCL contribute non-trivially to overall performance.

## 5.4 Varying Label Sizes in Fine-Tuning

We also assessed the effect of our pretraining on sample efficiency and learning efficiency during fine-tuning, especially when the amount of labels is limited. We changed the number of labeled samples from 2000 to 12000 during fine-tuning. As shown in Figure 4(c), the pretrained model consistently achieves lower RMSEs across all label sizes, with the most notable improvements observed with fewer labeled samples (lower mean RMSE and smaller variance). Moreover, pretraining appears to accelerate the convergence during the fine-tuning phase. To confirm this, we used an early stopping mechanism with a patience of 10 epochs. Figure 4(d) shows that the pretrained model not only starts with a lower initial RMSE but also reaches convergence more quickly (around 20 epochs) compared to the base model without our pretraining (over 40 epochs). This confirms that our pretraining enhances the label sample efficiency during fine-tuning.

## 5.5 Model Generalizability

It has been shown that the default split of general (training), refined (validation), and core (test) datasets in PDBbind is cross-contaminated with proteins and ligands with high similarity. To rigorously evaluate the effect of our pretraining on a model's generalizability, we used a leakage-proof split called LP-PDBbind [Li *et al.*, 2024a], which partitions PDBbind into training (10980 samples), validation (2312 samples), and test (4651 samples) sets based on high sequence and structural similarity across proteins and ligands. As a comparison, we also randomly split the data into training, validation, and test sets with the same sizes. The experimental results in Table 6 on the GIGN base model show a much more significant improvement after pretraining on the LP-PDBbind compared with a random split. This indicates that our pretraining framework enhances the generalizability of a base model across different protein and ligand structures.

Table 6: Model generalizability test on a leakage-proof split (LP-PDBbind)

|  | No pretrain | Pretrain |
|---|---|---|
| LP-PDBbind split | 1.496 (0.029) | 1.371 (0.006) |
| Random split | 1.294 (0.017) | 1.269 (0.015) |

## 5.6 Impact of Decoy Dataset Size and Quality

We assessed the impact of pretraining dataset (DecoyDB) configuration over model performance through: (1) varying the number of decoys by applying different ratios of negative pairs to positive pairs (negative-to-positive ratio); and (2) by filtering out the top 1% highest-RMSD decoys (outliers). We used the GIGN backbone for pretraining and PDBBind Core 2016 for fine-tuning. The results are from ten runs. As shown in Figure 5(a), performance improves by increasing the number of decoys and peaks at 10:1, after which the gain slightly fluctuates. We used 10:1 as the default in our main experiments. Filtering out extremely high-RMSD decoys makes little difference (Figure 5(b)).

# 6 Conclusion and Future Works

In this paper, we propose *DecoyDB*, a comprehensive dataset of high-resolution 3D complexes augmented with decoys for pretraining graph neural networks in protein-ligand binding affinity prediction. We also design a customized GCL algorithm based on DecoyDB. Experiments show that pretraining on DecoyDB improves multiple base models in prediction accuracy, sample learning efficiency, and model generalizability.

In the future, we plan to extend the proposed framework to other tasks beyond binding affinity prediction, such as binding pose estimation. Another interesting direction is to systematically study how the choice of decoy generation tools and their parameterizations influences model performance.

## Acknowledgments

This material is based upon work supported by the National Science Foundation (NSF) under Grant No. IIS-2147908, IIS-2207072, OAC-2152085, OAC-2402946, and OAC-2410884, the Bodor Professorship, as well as NIH R01CA212403 and R21EB037868.

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

# A  Experiment Setup

## A.1  Evaluation metrics

In this section, we provide the details about our two evaluation metrics in our experiment. Root Mean Square Error (RMSE) and Pearson's correlation coefficient (R) are denoted as:

$$RMSE = \sqrt{\frac{1}{N}\sum_{i=1}^{N}(y_i - \hat{y}_i)^2} \tag{6}$$

$$R = \frac{\sum_{i=1}^{N}(y_i - \bar{y})(\hat{y}_i - \bar{\hat{y}})}{\sqrt{\sum_{i=1}^{N}(y_i - \bar{y})^2 \sum_{i=1}^{N}(\hat{y}_i - \bar{\hat{y}})^2}} \tag{7}$$

$y_i$ and $\hat{y}_i$ represents experimental and predicted binding affinity, respectively.

## A.2  Data Preprocessing

The edges $E_k$ between atoms can be determined by the chemical bonds (covalent bonds) or a distance threshold. In our case, we used a distance threshold (5Å) to establish edges within the protein and the interactive edges between protein atoms and ligand atoms. We used the same distance threshold to filter out relevant atoms within protein binding pockets for the graphs. We kept the covalent bonds within a ligand as ligand edges.

## A.3  Training Parameters

In our experiments, we trained and evaluated the DeepDTA, GraphDTA, Pafnucy, OnionNet, SchNet, EGNN, TorchMD-Net and GIGN models. The Adam optimizer was employed for all models with the following settings: the initial learning rate was set to $5 \times 10^{-4}$, weight decay was set to $1 \times 10^{-6}$. To prevent overfitting and enhance the generalization ability of the models, we reduced the learning rate by a factor of 0.1 whenever the performance on the validation set did not improve for 10 consecutive epochs. Additionally, an early stopping strategy was adopted for all models, where training was terminated if the validation performance did not improve within 40 consecutive epochs. The pretraining phase for each model was set to 20 epochs, while the maximum number of epochs for the fine-tuning phase was set to 300. In the fine-tuning phase, we set the batch size to 128. In the pretraining phase, we set the batch size to 8. Each sample in the batch contained 21 structures: 1 real data sample, 10 randomly selected decoys, and 10 perturbed data samples generated by adding Gaussian noise. In addition to performance improvements, we also examined the time cost of our framework. Using GIGN as an example, the pretraining stage took approximately 23 hours, while the fine-tuning stage required only about 30 minutes.

## A.4  Model Architecture Parameters

In our experiments, we used the following parameter settings for the baseline models to ensure optimal performance. For Pafnucy, we set the channels of the three-layer 3D convolutions to 64, 128, and 256, respectively. For OnionNet, the input features were set to 3840, and there are 3 convolutional layers with 32, 64, and 128 filters. The kernel size was set to 4. The maximum length of protein sequences in DTA method was set to 1000. SchNet was configured with 3 interaction layers, each having an embedding dimension of 128. Both EGNN and GIGN models utilized a three-layer architecture with each layer having a dimension of 256. For TorchNet-MD, there are 6 layers and each layer has 256-dimensional embeddings.

# B  Baseline models

- **Docking Method**: **AutoDock Vina** [Eberhardt *et al.*, 2021] is a widely used program for molecular docking, which can also predict binding affinity. Since it is not a deep learning model, we directly cite the results from [Macari *et al.*, 2020]. Note that there are no repeated experiments, so there is no standard error being reported.

- **Drug-Target Affinity (DTA) methods: DeepDTA** [Lennox *et al.*, 2021] predict binding affinity based on protein and ligand sequences separately without interaction networks. **GraphDTA** [Nguyen *et al.*, 2020] is similar to DeepDTA but uses a GNN.
- **CNN-based methods: Pafnucy** [Stepniewska-Dziubinska *et al.*, 2018b] is a 3D-CNN model. **OnionNet** [Zheng *et al.*, 2019] uses 2D-CNN to learn representations.
- **GNN-based methods: TorchMD-Net** [Thölke and De Fabritiis, 2022] is a GNN-based model specially designed for the force field. **SchNet** [Schütt *et al.*, 2017] is a GNN model based on essential quantum chemical constraints. **EGNN** [Satorras *et al.*, 2021] is a GNN based on rotation and translation equivariance. **GIGN** [Yang *et al.*, 2023] is a GNN specifically designed for protein-ligand interactions. This is among the state of the art methods.
- **pretraining for general graphs:** General **GCL** based on edge perturbation and node dropping. We only tested these methods on top of the best baseline GNN model GIGN, including GCL with edge perturbation (GIGN+GCL-EP) or node dropping (GIGN+GCL-ND).
- **pretraining for biochemistry graphs:** (1) **Frad** [Ni *et al.*, 2024] designed a chemical-guided noise and utilizes denoising for pretraining. We only evaluated Frad on top of TorchMD-Net due to the availability of source codes. (2) **ConBAP** [Luo *et al.*, 2024] designs a contrastive learning strategy based on (binary) negative samples from decoys. We only evaluated ConBAP on EGNN due to the availability of source codes.
- **OURS**: This is our proposed graph contrastive learning framework. We evaluated our framework on top of GIGN, EGNN, and SchNet.

## C  Per-run Results

Table 7: R (2016) per-run results for SchNet vs SchNet + OURS.

| Run Index | Run 1 | Run 2 | Run 3 | Run 4 | Run 5 | Run 6 | Run 7 | Run 8 | Run 9 | Run 10 |
|---|---|---|---|---|---|---|---|---|---|---|
| SchNet | 0.742 | 0.741 | 0.717 | 0.762 | 0.761 | 0.747 | 0.755 | 0.740 | 0.747 | 0.728 |
| SchNet + OURS | 0.758 | 0.748 | 0.764 | 0.748 | 0.771 | 0.753 | 0.763 | 0.747 | 0.763 | 0.730 |

Table 8: RMSE (2013) per-run results for EGNN + OURS vs EGNN + ConBAP.

| Run Index | Run 1 | Run 2 | Run 3 | Run 4 | Run 5 | Run 6 | Run 7 | Run 8 | Run 9 | Run 10 |
|---|---|---|---|---|---|---|---|---|---|---|
| EGNN + ConBAP | 1.478 | 1.539 | 1.503 | 1.494 | 1.398 | 1.502 | 1.459 | 1.485 | 1.499 | 1.436 |
| EGNN + OURS | 1.473 | 1.436 | 1.460 | 1.399 | 1.502 | 1.484 | 1.400 | 1.354 | 1.459 | 1.406 |

## D  Pseudocode

Our framework is in Algorithm 1 and Algorithm 2. For simplicity, we omit the loop over epochs.

---
**Algorithm 1** Graph Contrastive Pretraining
---
1: **Input:** Protein-ligand complexes $\mathcal{S}$, a GNN encoder $f_\theta$, *DecoyDB*
2: **Output:** Pretrained parameters $\theta$ of encoder $f_\theta$
3: **for** each unlabeled protein-ligand complex $s \in \mathcal{S}$ **do**
4:     With $s$ as an anchor, sample positive and negative pairs from *DecoyDB*
5:     Compute latent representation $\mathbf{z} = f_\theta(s)$ for them
6:     Compute contrastive loss $L_1$ (Equation 3)
7:     Sample Gaussian noise $\epsilon$
8:     Apply noise to ligand coordinates: $\mathbf{x}' = \mathbf{x} + \epsilon$
9:     Compute denoising loss $L_2$ (Equation 4)
10:    Compute total loss: $\mathcal{L}$ (Equation 5)
11:    Update encoder parameters: $\theta \leftarrow \theta - \eta \nabla_\theta \mathcal{L}$
12: **end for**
---

---

**Algorithm 2** Fine-tuning

---

1: **Input:** Pretrained encoder $f_\theta$, labeled protein-ligand complexes $S_l$ and a new regression head $g_\phi$
   s.t. $\hat{y} = g_\phi(f_\theta(s))$
2: **Output:** Fine-tuned parameters $\theta$ and $\phi$
3: Replace regression head $g_\phi$
4: **for** each labeled protein-ligand complex $s \in S_l$ **do**
5:     Predict binding affinity: $\hat{y} = g_\phi(f_\theta(s))$
6:     Compute loss: $\mathcal{L}_{\text{finetune}} = (\hat{y} - y)^2$
7:     Update parameters: $[\theta, \phi] \leftarrow [\theta, \phi] - \eta \nabla_{\theta, \phi} \mathcal{L}_{\text{finetune}}$
8: **end for**

---

