# OpenReview forum: "DecoyDB: A Dataset for Graph Contrastive Learning in Protein-Ligand Binding Affinity Prediction"
_NeurIPS.cc/2025/Datasets_and_Benchmarks_Track — NeurIPS 2025 Datasets and Benchmarks Track poster_

### Official Review · Reviewer_hDDp · 2025-06-30

**Rating:** 5
**Confidence:** 2

**Summary:**

The authors present DecoyDB, a dataset of artificially created unlabeled suboptimal protein-ligand complexes created from optimal, labeled protein-ligand complexes in the PDBbind database.
The newly created protein-ligand complexes deviate by a varying amount from the natural pose and a deviation score can be computed for them. This allows to use the DecoyDB for self-supervised contrastive learning as a first step to improve the supervised affinity prediction problem on the PDBbind or other similar databases.
The empirical results show improvements over direct supervised training on only PDBbind.

**Dataset Code Accessibility:**

Yes

**Dataset Code Comments:**

The code is hosted on github, the dataset is hosted on huggingface.

**Ethical Comments:**

The paper discusses data augmentation for the improvement of affinity prediction of protein-ligand complexes. I see no ethical concerns here.

**Ethical Considerations:**

No, there are no or only very minor ethics concerns

**Final Justification:**

The authors present a principled and chemically valid way to generate graphs that can be considered 'close' and 'far' from a given experimentally confirmed protein-ligand complex. In this way, the generated data is much more realistic than the usual 'ad-hoc' random augmentations that are used to generate augmentations of given graphs for contrastive learning.

The rebuttal of the authors did not raise any novel issues that prompted me to change my score.

**Limitations Weaknesses:**

The practical relevance of the proposed pretraining depends on the choice of method (Autodock) and parameters to compute the decoys. The paper does not investigate the influence of those on the overall process. A discussion could improve the paper.

**Strengths Contributions:**

The authors present a principled and chemically valid way to generate graphs that can be considered 'close' and 'far' from a given experimentally confirmed protein-ligand complex. In this way, the generated data is much more realistic than the usual 'ad-hoc' random augmentations that are used to generate augmentations of given graphs for contrastive learning.

---

> ### Author Rebuttal · Authors · 2025-07-31
>
> Thank you for your positive feedback and for highlighting the significance of our work.
>
> **W1. Decoy generation method and parameters:** We appreciate your suggestion. Exploring the impact of different docking tools (e.g., AutoDock Vina) and parameters (e.g., exhaustiveness, number of poses, scoring functions) is indeed valuable. Since widely used docking programs typically generate a high proportion of physically plausible poses, the resulting decoys may share similar characteristics. While a detailed empirical comparison is beyond the scope of this work, we agree it is an important direction and have included it in the Future Work section.

---

> > ### Comment · Reviewer_hDDp · 2025-08-04
> >
> > Thank you for your answer. I maintain my score

---

### Official Review · Reviewer_PcXA · 2025-07-01

**Rating:** 4
**Confidence:** 3

**Summary:**

This paper proposes and constructs a large-scale, structure-aware protein-ligand complex dataset called DecoyDB, specifically designed for self-supervised graph contrastive learning (GCL) in protein-ligand binding affinity prediction. DecoyDB includes high-resolution ground truth complexes and computationally generated decoy structures, ranging from realistic to suboptimal (negative pairs), with each decoy annotated with its Root Mean Squared Deviation (RMSD) from the native pose. The authors also designed a customized GCL framework that combines a two-category contrastive loss and denoising score matching (DSM)-based regularization for pre-training graph neural networks. Experimental results claim that models pre-trained with DecoyDB achieve superior accuracy, label efficiency, and generalizability.

**Dataset Code Accessibility:**

Yes

**Ethical Considerations:**

No, there are no or only very minor ethics concerns

**Limitations Weaknesses:**

1. RThe paper uses RMSD to quantify the "negativeness" of decoys and weight continuous negative samples. However, RMSD alone may not fully represent the suboptimal nature of binding energy or affinity. High-RMSD conformations could still have plausible binding modes, while low-RMSD ones might differ significantly in affinity due to subtle local interactions. The paper does not deeply explore the complex relationship between RMSD and actual binding free energy or energy landscapes.
2. The paper lacks citations for some relevant prior works, e.g. [1].
3. Although ablation studies show performance improvements from both GCL and DSM modules , the "modest increase" in RMSE improvement is insufficient to fully quantify each module's specific contribution or synergistic effects.

[1]Koh H Y, Nguyen A T N, Pan S, et al. Physicochemical graph neural network for learning protein–ligand interaction fingerprints from sequence data[J]. Nature Machine Intelligence, 2024, 6(6): 673-687.

**Strengths Contributions:**

1. DecoyDB includes 61,104 real 3D complexes and 5,353,307 decoys, with an average of 88 decoys per complex. RMSD values range from 0.03Å to 25.56Å, averaging 7.22Å, offering diverse binding interactions and spatial conformations for learning.
2. Unlike methods using simple graph perturbations, DecoyDB's computationally generated decoys preserve biochemical constraints and structural information, preventing unrealistic conformations.
3. The paper introduces a customized GCL framework with a two-category contrastive loss (considering continuous decoy negative samples and discrete negative samples from different real complexes) and denoising score matching (DSM) regularization. This design accounts for the degree of "negativeness" in samples and aims to capture local minimum energy characteristics.

---

> ### Author Rebuttal · Authors · 2025-07-31
>
> We thank the reviewer for the thoughtful feedback and hope our responses clarify the concerns.
>
> **W1. RMSD to represent binding affinity:** Thank you for raising this important point. We agree that RMSD does not fully capture binding affinity or the underlying energy landscape. Our use of RMSD is not meant to approximate binding energy or affinity, but rather as a geometric measure of how closely a decoy resembles the ground truth crystal structure. It provides a practical and natural criterion for defining positive and negative pairs during contrastive learning in the pretraining stage, helping the model learn meaningful geometric representations of binding poses. It is not the intention of this paper to "deeply explore the complex relationship between RMSD and actual binding free energy or energy landscapes". We have clarified this distinction in the revised manuscript.
>
> **W2. Missing citation:** Thank you for pointing this out. We have added the suggested citation to the Related Work section in the revised manuscript.
>
> **W3. Fully quantifying module contribution:** Thank you for the constructive feedback. To further assess the contribution of each module, we performed paired t-tests with 10 independent runs for all model variants. The results show that the RMSE improvements are statistically significant (p < 0.05) compared to their ablated counterparts, indicating that each component contributes non-trivially to overall performance.
>
> | Ablation Comparison              | p-value      |
> |----------------------------------|--------------|
> | No pretraining vs Ours           | 0.0280       |
> | Ours w/o DSM vs Ours             | 0.0013       |
> | Ours w/o GCL vs Ours             | 0.0325       |
>
> These results provide stronger evidence that the observed improvements are not due to random variation, but to the specific design of the proposed modules. We have included this analysis in the updated manuscript.

---

### Official Review · Reviewer_ecxx · 2025-07-02

**Rating:** 4
**Confidence:** 4

**Summary:**

This paper constructs DecoyDB, a large-scale protein-ligand complex dataset containing both real-world data and diverse decoy data. DecoyDB contains over 5 million decoys generated using AutoDock Vina and annotated with RMSD from native poses, enabling the definition of structured positive/negative pairs. Based on the dataset, a pre-training framework is also proposed, which includes two training objectives: two-category Graph Contrastive Learning (GCL) between positive/negative pairs, and Denoising Score Matching (DSM) loss for regularization. Experimental results show that models pre-trained on DecoyDB consistently outperform baselines, highlighting the value of this large-scale decoy dataset.

**Dataset Code Accessibility:**

Yes

**Dataset Code Comments:**

The codes and datasets are available.

**Ethical Considerations:**

No, there are no or only very minor ethics concerns

**Final Justification:**

The reviewer's concerns are fully addressed, hence the scores are kept for acceptance.

**Limitations Weaknesses:**

1. Since pretraing on the decoy dataset is a key contribution, it would be valuable to analyze how the size of the decoy data influence downstream performance. For example, further investigation could explore how varying the ratio of negative samples impacts the results.

2. DecoyDB contains negative samples with a wide range of RMSDs. It would be useful to investigate whether excluding decoys with excessively high RMSD could improve the pretrained models.

3. It remains unclear whether the DSM loss is applied only to ground-truth complexes or also to decoys. Clarifying this point would improve the understanding of the pretraining process.

**Strengths Contributions:**

1. The proposed dataset fills the gap in current domain by offering a large-scale synthesized decoy dataset.

2. Apart from the dataset, this paper also designs a poweful pretraining framework, which is technically sound and provides a strong foundation for future research.

3. The paper benchmarks a wide range of baselines across multiple families, making the evaluations relatively comprehensive.

4. The datasets and the training codes are openly released.

---

> ### Author Rebuttal · Authors · 2025-07-31
>
> We thank the reviewer for the time and effort dedicated to reviewing our work. Below are our responses:
>
> **W1. Analysis of decoy data size:** Thank you for the insightful suggestion. We agree that the size of the decoy data and the ratio of negative to positive samples during pretraining is an important factor. Following your recommendation, we conducted additional experiments varying this ratio (2:1, 4:1, 6:1, 10:1, 12:1, 14:1, 16:1):
>
>
> | Negative Ratio (Neg:Pos) | 2:1            | 4:1            | 6:1            | 10:1           |12:1            | 14:1       |16:1|
> |--------------------------|----------------|----------------|----------------|----------------|----------------|------       |-----|
> | RMSE                     | 1.266 (0.020)  | 1.250 (0.029)  | 1.202 (0.018)  | 1.188 (0.022)  | 1.200 (0.009)  | 1.198 (0.016)| 1.190 (0.021)|
>
> As shown, performance improves as the ratio increases up to 10:1. Using more negative samples beyond that does not help or even can slightly hurt performance. We use the 10:1 ratio in our main experiments.
>
>
> **W2. Excluding decoys with excessively high RMSD:** To evaluate the effect of high-RMSD decoys, we conducted an additional experiment where we filtered out the top 1% of decoys with the highest RMSD during pretraining. Due to time constraints, this was performed using the GIGN model only.
>
> | Model               | RMSE (↓)       | R (↑)         |
> |--------------------|----------------|---------------|
> | Full set    | 1.188 (0.022)  | 0.840 (0.008) |
> | Filtered set  | 1.197 (0.024)  | 0.838 (0.011) |
>
> As shown, filtering out extreme high-RMSD decoys results in only minor changes in performance. This suggests that such outliers have limited impact on downstream results.
>
>
> **W3. DSM loss:** Thank you for pointing this out. We clarify that the DSM loss is **only applied to ground-truth** complexes, not to decoys. We have updated the manuscript to make this explicit.

---

> > ### Comment · Reviewer_ecxx · 2025-08-09
> >
> > Thanks for the detailed responses! My concerns are fully addressed and I would like to keep my original score.

---

### Official Review · Reviewer_nS73 · 2025-07-03

**Rating:** 4
**Confidence:** 3

**Summary:**

This paper proposes DecoyDB, a dataset for protein-ligand binding affinity prediction with graph contrastive learning (GCL) methods, which features 61k protein-ligand complexes with 5 million computationally generated decoys. The authors then introduced a new contrastive learning framework based on InfoNCE with an additional denoising regularization term, which is later tested on the proposed dataset. In the experiments, the authors compared the performance of their methods to other GCL baselines by first pretraining on DecoyDB or other existing benchmarks, then finetuning on the PDBbind dataset and comparing the testing RMSE & Pearson’s correlation coefficient.

**Additional Feedback:**

How is the threshold (2Å) of the positive/negative pair determined? Based on Figure2 (d), almost all decoys are negative pairs, could you explain the rationale behind this decision?

**Dataset Code Accessibility:**

Yes

**Dataset Code Comments:**

The dataset is hosted on GitHub and HuggingFace, where the authors also provide a README.md file to run their code.

**Ethical Considerations:**

No, there are no or only very minor ethics concerns

**Final Justification:**

The authors' rebuttal addressed my concerns, and I updated my score to 4 accordingly.

**Limitations Weaknesses:**

My primary concern with this work is the empirical performance, as shown in Table 2, where the differences between their proposed method and the baselines are statistically insignificant in many cases.

To facilitate discussion, I append a copy of the GNN part from Table 2 below, which shows the performance comparison on PDBbind core set 2013 and core set 2016. In addition, I also **highlight** the results that are **insignificantly different** from the proposed method (OURS & DecoyDB). Note that I did not run the two-sample t-test; however, if the difference is within 1 standard deviation of either method, I believe it is a clear signal of insignificance.

| Method           | Pre-training dataset | RMSE (2013) ↓ | R (2013) ↑ | RMSE (2016) ↓ | R (2016) ↑ |
|------------------|----------------------|---------------|------------|---------------|------------|
|           SchNet           | –        | **1.662 ± 0.074** | 0.727 ± 0.030 | **1.471 ± 0.019** | **0.758 ± 0.017** |
|           *+ OURS*           | *DecoyDB*  | *1.626 ± 0.056* | *0.763 ± 0.026* | *1.434 ± 0.051* | *0.779 ± 0.024* |
|------------------|----------------------|---------------|------------|---------------|------------|
|           EGNN             | –        | 1.508 ± 0.046 | **0.774 ± 0.019** | 1.304 ± 0.033 | 0.807 ± 0.014 |
|           + ConBAP         | Redocked 2020 | **1.427 ± 0.047** | **0.786 ± 0.014** | 1.285 ± 0.022 | **0.814 ± 0.016** |
|           + *OURS*           | *DecoyDB*  | *1.417 ± 0.028* | *0.790 ± 0.013* | *1.250 ± 0.017* | *0.817 ± 0.022* |
|------------------|----------------------|---------------|------------|---------------|------------|
|           GIGN             | –        | 1.460 ± 0.050 | **0.787 ± 0.024** | 1.263 ± 0.020 | 0.814 ± 0.006 |
|           + GCL-EP         | DecoyDB  | 1.455 ± 0.006 | **0.794 ± 0.021** | 1.283 ± 0.015 | 0.812 ± 0.011 |
|           + GCL-ND         | DecoyDB  | 1.467 ± 0.016 | 0.700 ± 0.008 | 1.268 ± 0.043 | 0.819 ± 0.006 |
|           + *OURS*           | *DecoyDB*  | *1.386 ± 0.043* | *0.801 ± 0.011* | *1.188 ± 0.022* | *0.840 ± 0.008* |


We can observe that many baseline results are within 1 standard deviation of the results from the proposed method. In particular, while the result of EGNN pretrained on Redocked is insignificant from OURS+DecoyDB, the size of Redocked is < 20% of DecoyDB according to Table 1 in the paper. Does this imply a superior data quality in Redocked compared to the proposed dataset?


I understand that ignoring the significance test is somewhat “acceptable” in research papers at machine learning venues, where the contribution of one work depends not only on the performance, but also on the theoretical and methodological insights. Nevertheless, in the current benchmark paper, I believe the contribution of the proposed dataset and the new GCL method is more on the empirical side.

With that being said, I still believe this work has a good contribution to the field, and I am open to further discussion with the authors.

**Strengths Contributions:**

(1) This paper is generally well-written, where the motivation is clear, the task is properly defined, and the dataset construction and the proposed new GCL method are easy to understand.

(2) The review of literature is adequate. The authors discussed the commonly used benchmarks for protein-ligand affinity prediction, as well as the latest works on GCL methods.

---

> ### Author Rebuttal · Authors · 2025-07-31
>
> Thank you very much for your valuable review; it is crucial for improving the quality of our manuscript.
>
> **W1. Significance of results:** Thank you for your thoughtful and constructive comments. We fully agree that statistical significance is critical, and we appreciate your careful reexamination of the results in Table 2. To better address this concern, we have extended our experiments:
> - We increased the number of repeated runs from 3 to 10 across all model variants to reduce variance.
>
> | **Model**        | **RMSE (2013)** | **R (2013)** | **RMSE (2016)** | **R (2016)** |
> |------------------|---------------------|------------------|---------------------|------------------|
> | SchNet                                    | 1.642 (0.030)       | 0.739 (0.016)    | 1.526 (0.037)       | 0.744 (0.014)    |
> |  + OURS                                   | 1.577 (0.034)      | 0.763 (0.015)    | 1.481 (0.031)       | 0.755 (0.012)    |
> | EGNN                                      | 1.496 (0.047)       | 0.761 (0.012)    | 1.334 (0.024)       | 0.801 (0.010)    |
> | + ConBAP (Redocked 2020)                  | 1.479 (0.038)       | 0.766 (0.010)    | 1.300 (0.019)       | 0.802 (0.014)    |
> | + OURS                                    | 1.437 (0.044)       | 0.781 (0.013)    | 1.267 (0.021)       | 0.813 (0.011)    |
> | GIGN                                      | 1.421 (0.038)       | 0.786 (0.016)    | 1.262 (0.032)       | 0.811 (0.010)    |
> | + GCL-EP                                  | 1.417 (0.034)       | 0.789 (0.014)    | 1.251 (0.029)       | 0.814 (0.008)    |
> | + GCL-ND                                  | 1.420 (0.026)       | 0.787 (0.011)    | 1.254 (0.026)       | 0.813 (0.013)    |
> | + OURS                                    | 1.377 (0.039)       | 0.813 (0.013)    | 1.189 (0.031)       | 0.838 (0.011)    |
>
> - We computed paired t-tests for all key comparisons to formally assess statistical significance.
>
> | **Comparison**                    | **RMSE (2013)** | **R (2013)**  | **RMSE (2016)**| **R (2016)**   |
> |-----------------------------------|-----------------|-------------- |----------------|----------------|
> | SchNet + OURS vs SchNet           | 0.000563        | 0.003835      | 0.000151       | 0.069592       |
> | EGNN + OURS vs EGNN               | 0.040267        | 0.000691      | 0.000861       | 0.034137       |
> | EGNN + OURS vs EGNN + ConBAP      | 0.071769        | 0.000434      | 0.045742       | 0.022523       |
> | GIGN + OURS vs GIGN               | 0.015525        | 0.001407      | 0.000186       | 0.000046       |
> | GIGN + OURS vs GIGN + GCL-EP      | 0.019465        | 0.004697      | 0.000124       | 0.000650       |
> | GIGN + OURS vs GIGN + GCL-ND      | 0.011231        | 0.003200      | 0.000142       | 0.000360       |
>
> The results show that performance improvements are statistically significant (p < 0.05 or p < 0.01) in most comparisons, including both RMSE and R on 2013 and 2016 dataset, while a few comparisons fall short of conventional thresholds (e.g., EGNN+OURS vs ConBAP in RMSE 2013).
>
> - We also provide per-run results for specific cases where the statistical difference is not significant.
>
> **R (2016) for SchNet vs SchNet + OURS:**
>
> | Run Index      | Run 1  | Run 2  | Run 3  | Run 4  | Run 5  | Run 6  | Run 7  | Run 8  | Run 9  | Run 10 |
> |----------------|--------|--------|--------|--------|--------|--------|--------|--------|--------|--------|
> | SchNet         | 0.742  | 0.741  |  0.717 | 0.762  |  0.761 |  0.747 | 0.755  | 0.740  | 0.747  | 0.728  |
> | SchNet + OURS  | 0.758  | 0.748  | 0.764  | 0.748  | 0.771  | 0.753  |  0.763 | 0.763  | 0.747  | 0.730  |
>
>
> **RMSE (2013) for EGNN + OURS vs EGNN + ConBAP:**
>
> | Run Index      | Run 1  | Run 2  | Run 3  | Run 4  | Run 5  | Run 6  | Run 7  | Run 8  | Run 9  | Run 10  |
> |----------------|--------|--------|--------|--------|--------|--------|--------|--------|--------|---------|
> | EGNN + ConBAP  | 1.478 | 1.539 | 1.503 | 1.494 | 1.398 | 1.502 | 1.459 | 1.485 | 1.499  | 1.436  |
> | EGNN + OURS    | 1.473 | 1.436 | 1.460  | 1.399 | 1.502 |  1.484 |1.400  |  1.354 | 1.459 | 1.406 |
>
>
>
>
> From the above 2 tables, we observe that although the improvements brought by our method do not always reach statistical significance under the standard t-test threshold (p < 0.05), our approach consistently outperforms the baselines across most random splits.
>
> Lastly, we would like to emphasize that the primary contribution of this work lies in introducing a new dataset and pretraining paradigm. Our goal is to establish reasonable and reproducible baselines that demonstrate the utility of DecoyDB, rather than to fully optimize its usage. We hope this dataset will support future research by providing a large-scale, physically grounded source of decoy data and enabling new directions in self-supervised learning for this domain.
>
> **W2. DecoyDB vs Redocked 2020:** Thank you for raising this point. We believe it's difficult to assess the “quality” of DecoyDB and Redocked 2020 purely based on dataset size or performance. These datasets represent fundamentally different approaches to decoy generation. Redocked 2020 redocks ligands into protein conformations, creating synthetic protein-ligand pairs through docking. In contrast, DecoyDB perturbs experimentally resolved protein-ligand complexes, generating synthetic poses from real complexes, which may lead to more physically grounded and realistic decoys. A comprehensive comparison between the two strategies would require further investigation, which we can leave for future work.
>
>
> **W3. Choice of RMSD threshold:** Thank you for the question. While there is no fixed rule, an RMSD threshold of 2 Å is a widely adopted and recommended cutoff for assessing docking success with regular-sized ligands, as supported by prior work [1,2]. It reflects the standard for defining near-native binding poses in the molecular docking community. Additionally, the crystal structures used as ground truth in our study have resolutions of ≤ 2.5 Å, so setting the threshold at 2 Å ensures that positive samples fall within the structural precision of the reference data. As shown in Figure 2(d), most decoys are indeed structurally distant from the native pose and are thus treated as negatives under this criterion. We have clarified this rationale in the revised manuscript.
>
> [1] Buttenschoen, Martin, et al. *"PoseBusters: AI-based docking methods fail to generate physically valid poses or generalise to novel sequences."* Chemical Science 15.9 (2024): 3130-3139.
> [2] Cole, Jason C., et al. *"Comparing protein–ligand docking programs is difficult."* Proteins: Structure, Function, and Bioinformatics 60.3 (2005): 325–332.

---

> > ### Comment · Reviewer_nS73 · 2025-08-01
> >
> > I would like to thank the authors for their rebuttal. I have no further questions and will update my score to the positive.

---

> > > ### Author Response · Authors · 2025-08-01
> > >
> > > Thank you very much for your kind feedback. We sincerely appreciate your time and effort in reviewing our paper.

---

### Decision · Program_Chairs · 2025-09-18

**Decision:**

Accept (poster)

**Comment:**

The manuscript and related supplemental materials present a new large scale dataset for prediction of protein-ligand binding affinity with structure-aware graph contrastive learning. The scale of the data is impressive, containing a mix of real-world and computationally generated (decoy) entries. Reviewers pointed out positively not only the scale, but also the realistic nature of the provided data, as well as the pretraining capabilities enabled by it. Some concerns were raised, for example, about the significance of the data for benchmarking performance (e.g., beyond statistical significance), but these were addressed in rebuttal. All the reviewer concerns seem to have been sufficiently addressed in the rebuttal, and the potential revisions expected to address them in the camera-ready version are well within reason. Following the rebuttal period, all reviews are positive (one accept, the rest borderline accept), and all seem satisfied with accepting the paper, and therefore I confidently recommend it be accepted.

Furthermore, given the large scale and complexity of the data, its realistic chemically well motivated nature, and the importance of this domain (including enabling self-supervised pretraining), I am comfortable recommending spotlighting this work to increase its exposure and attract researchers working on relevant methods to use it as a benchmark for their computational work.

===== FINAL UPDATE FROM DB Track PCs ====

The final decision for this paper has been taken by the program chairs after consultation with the SACs. All Senior Area Chairs have ranked papers according to the feedback from the AC during the review process. We decided to leave the original meta-review to reflect the opinion of the AC in light of the initial discussions with reviewers and SAC.